# NEURAL SINKHORN GRADIENT FLOW

## ABSTRACT

Wasserstein Gradient Flows (WGF) with respect to specific functionals have been widely used in the machine learning literature. Recently, neural networks have been adopted to approximate certain intractable parts of the underlying Wasserstein gradient flow and result in efficient inference procedures. In this paper, we introduce the Neural Sinkhorn Gradient Flow (NSGF) model, which parametrizes the time-varying velocity field of the Wasserstein gradient flow w.r.t. the Sinkhorn divergence to the target distribution starting a given source distribution. We utilize the velocity field matching training scheme in NSGF, which only requires samples from the source and target distribution to compute an empirical velocity field approximation. Our theoretical analyses show that as the sample size increases to infinity, the mean-field limit of the empirical approximation converges to the true underlying velocity field. With specific source and target data samples, our NSGF models can be used in many machine learning tasks such as unconditional/conditional image generating, style transfer, and audio-text translations. Numerical experiments with synthetic and real-world benchmark datasets support our theoretical results and demonstrate the effectiveness of the proposed method.

## 1 INTRODUCTION

The Wasserstein Gradient Flow (WGF) with respect to certain specific functional objective $\mathcal{F}$ (denoted as $\mathcal{F}$ Wasserstein gradient flow) is a powerful tool for solving optimization problems over the Wasserstein probability space. Since the seminal work of Jordan et al. (1998) which shows that the Fokker-Plank equation is the Wasserstein gradient flow with respect to the free energy, Wasserstein gradient flow w.r.t. different functionals have been widely used in various machine learning tasks such as Bayesian inference (Liu, 2017; Liu et al., 2019; di Langosco et al., 2021; Zhang et al., 2021a), reinforcement learning (Zhang et al., 2018; Martin et al., 2020; Agazzi & Lu, 2020; Zhang et al., 2021b), and mean-field games (Domingo-Enrich et al., 2020; Gao et al., 2022b; Zhang & Katsoulakis, 2023).

One recent trend in the Wasserstein gradient flow literature is to develop efficient generative modeling methods (Gao et al., 2019; 2022a; Ansari et al., 2021; Mokrov et al., 2021; Alvarez-Melis et al., 2022; Bunne et al., 2022; Fan et al., 2022). In general, these methods mimic the Wasserstein gradient flow with respect to a specific distribution metric, driving a source distribution towards a target distribution. Neural networks are typically employed to approximate the computationally challenging components of the underlying Wasserstein gradient flow such as the time-dependent transport maps. During the training process of these methods, it is common to require samples from the target distribution. After the training process, an inference procedure is often employed to generate new samples from the target distribution This procedure involves iteratively transporting samples from the source distribution with the assistance of the trained neural network. Based on the chosen metric, these methods can be categorized into two main types.

**Divergences Between Distributions With Exact Same Supports.** The first class of widely used metrics is the f-divergence, such as the Kullback-Leibler divergence and the Jensen-Shannon divergence. These divergences are defined based on the density ratio between two distributions and are only well-defined when dealing with distributions that have exactly the same support. Within the scope of f-divergence Wasserstein gradient flow generative models, neural networks are commonly utilized to formulate density-ratio estimators, as demonstrated by Gao et al. (2019; 2022a); Ansari et al. (2021) and Heng et al. (2022). However, as one can only access finite samples from target distributions in the training process, the support shift between the sample collections from

the compared distributions may cause significant approximation error in the density-ratio estimators (Rhodes et al., 2020; Choi et al., 2022). An alternative approach, proposed by Fan et al. (2022), circumvents these limitations by employing a dual variational formulation of the f-divergence. In this framework, two networks are employed to approximate the optimal variational function and the transport maps. These two components are optimized alternately. It's imperative to highlight that the non-convex and non-concave characteristics of their min-max objective can render the training inherently unstable (Arjovsky & Bottou, 2017; Hsieh et al., 2021).

**Divergences Between Distributions With Possible Different Supports.** Another type of generative Wasserstein gradient flow model employs divergences that are well-defined for distributions with possible different supports. This includes free energy fuctionals (Mokrov et al., 2021; Alvarez-Melis et al., 2022; Bunne et al., 2022), the kernel-based metrics such as the Maximum-Mean/Sobolev Discrepancy (Mroueh et al., 2019; Mroueh & Rigotti, 2020; Altekrüger et al., 2023) and sliced-Wasserstein distance (Liutkus et al., 2019; Du et al., 2023). As these divergences can be efficiently approximated with samples, neural networks are typically used to directly model the transport maps used in the inference procedure. In Wasserstein gradient flow methods, input convex neural networks (ICNNs, Amos et al. (2017)) are commonly used to approximate the transport map. However, recently, several works (Bonet et al., 2021; Korotin et al., 2021) discuss the poor expressiveness of ICNNs architecture and show that it would result in poor performance in high-dimension applications. Besides, the Maximum-Mean/Sobolev discrepancy Wasserstein gradient flow models are usually hard to train and are easy to trapped in poor local optima in practice (Arbel et al., 2019), since the kernel-based divergences are highly sensitive to the parameters of the kernel function (Li et al., 2017; Wang et al., 2018). Liutkus et al. (2019); Du et al. (2023) consider sliced-Wasserstein WGF to build nonparametric generative Models which do not achieve high generation quality, it is an interesting work on how to combine sliced-Wasserstein WGF and neural network methods.

**Contribution.** In this paper, we investigate the Wasserstein gradient flow with respect to the Sinkhorn divergence, which is categorized under the second type of divergences and does not necessitate any kernel functions. We introduce the *Neural Sinkhorn Gradient Flow (NSGF)* model, which parametrizes the time-varying velocity field of the Sinkhorn Wasserstein gradient flow from a specified source distribution. The NSGF employs a velocity field matching scheme that demands only samples from the target distribution to calculate empirical velocity field approximations. Our theoretical analyses show that as the sample size approaches infinity, the mean-field limit of the empirical approximation converges to the true velocity field of the Sinkhorn Wasserstein gradient flow. Given distinct source and target data samples, our NSGF can be harnessed across a wide range of machine learning applications, including unconditional/conditional image generation, style transfer, and audio-text translation. We empirically validate NSGF on low-dimensional 2D data and benchmark images (MNIST, CIFAR-10). Our findings indicate that our models can be trained to yield commendable results in terms of generation cost and sample quality, surpassing the performance of the neural Wasserstein gradient flow methods previously tested on CIFAR-10, to the best of our knowledge.

## 2 RELATED WORKS

**Sinkhorn Divergence in Machine Learning.** Originally introduced in the domain of optimal transport, the Sinkhorn divergence emerged as a more computationally tractable alternative to the classical Wasserstein distance (Cuturi, 2013; Peyré et al., 2017; Feydy et al., 2019). Since its inception, Sinkhorn divergence has found applications across a range of machine learning tasks, including domain adaptation (Courty et al., 2014; Alaya et al., 2019; Komatsu et al., 2021), Sinkhorn barycenter (Luise et al., 2019; Shen et al., 2020) and color transfer (Blondel et al., 2018; Pai et al., 2021). Indeed, it has already been extended to single-step generative modeling methods, such as the Sinkhorn GAN and VAE (Genevay et al., 2018; Deja et al., 2020; Patrini et al., 2020). However, to the best of our knowledge, it has yet to be employed in developing efficient generative Wasserstein gradient flow models.

**Neural ODE/SDE Based Diffusion Models.** Recently, diffusion models, as a class of Neural ODE/SDE Based generative methods have achieved unprecedented success, which also transforms a simple density to the target distribution, iteratively (Song & Ermon, 2019; Ho et al., 2020; Song et al., 2021). Typically, each step of diffusion models only progresses a little by denoising a sim-

ple Gaussian noise, while each step in WGF models follows the most informative direction (in a certain sense). Hence, diffusion models usually have a long inference trajectory. In recent research undertakings, there has been a growing interest in exploring more informative steps within diffusion models. Specifically, flow matching methods (Lipman et al., 2023; Liu et al., 2023; Albergo & Vanden-Eijnden, 2023) establish correspondence between the source and target via optimal transport, subsequently crafting a probability path by directly linking data points from both ends. Notably, when the source and target are both Gaussians, their path is actually a Wasserstein gradient flow. However, this property does not consistently hold for general data probabilities. Moreover, Tong et al. (2023); Pooladian et al. (2023) consider calculating the minibatch optimal transport map to guide data points connecting. Besides, Das et al. (2023) consider the shortest forward diffusion path for the Fisher metric and Shaul et al. (2023) explore the conditional Gaussian probability path based on the principle of minimizing the Kinetic Energy. Nonetheless, a commonality among many of these methods is their reliance on Gaussian paths for theoretical substantiation, thereby constraining the broader applicability of these techniques within real-world generative modeling.

## 3 PRELIMINARIES

### 3.1 NOTATIONS

We denote $\boldsymbol{x} = (x_1, \cdots, x_d) \in \mathbb{R}^d$ and $\mathcal{X} \subset \mathbb{R}^d$ as a vector and a compact ground set in $\mathbb{R}^d$, respectively. For a given point $\boldsymbol{x} \in \mathcal{X}$, $\|\boldsymbol{x}\|_p := (\sum_i x_i^p)^{\frac{1}{p}}$ denotes the $p$-norm on euclidean space, and $\delta_{\boldsymbol{x}}$ stands for the Dirac (unit mass) distribution at point $\boldsymbol{x} \in \mathcal{X}$. $\mathcal{P}_2(\mathcal{X})$ denotes the set of probability measures on $\mathcal{X}$ with finite second moment and $\mathcal{C}(\mathcal{X})$ denotes the space of continuous functions on $\mathcal{X}$. For a given functional $\mathcal{F}(\cdot) : \mathcal{P}_2(\mathcal{X}) \to \mathbb{R}$, $\frac{\delta \mathcal{F}(\mu_t)}{\delta \mu}(\cdot) : \mathbb{R}^d \to \mathbb{R}$ denotes its first variation at $\mu = \mu_t$. Besides, we use $\nabla$ and $\nabla \cdot ()$ to denote the gradient and the divergence operator, respectively.

### 3.2 WASSERSTEIN DISTANCE AND SINKHORN DIVERGENCE

We first introduce the background of Wasserstein distance. Given two probability measures $\mu, \nu \in \mathcal{P}_2(\mathcal{X})$, the $p$-Wasserstein distance $\mathcal{W}_p(\mu, \nu) : \mathcal{P}_2(\mathcal{X}) \times \mathcal{P}_2(\mathcal{X}) \to \mathbb{R}_+$ is defined as:

$$\mathcal{W}_p(\mu, \nu) = \inf_{\pi \in \Pi(\mu, \nu)} \left( \int_{\mathcal{X} \times \mathcal{X}} \|\boldsymbol{x} - \boldsymbol{y}\|^p \, \mathrm{d}\pi(\boldsymbol{x}, \boldsymbol{y}) \right)^{\frac{1}{p}}, \tag{1}$$

where $\Pi(\mu, \nu)$ denotes the set of all probability couplings $\pi$ with marginals $\mu$ and $\nu$. The $\mathcal{W}_p$ distance aims to find a coupling $\pi$ so as to minimize the cost function $\|\boldsymbol{x} - \boldsymbol{y}\|^p$ of moving a probability mass from $\mu$ to $\nu$. It has been demonstrated that the $p$-Wasserstein distance is a valid metric on $\mathcal{P}_2(\mathcal{X})$, and $(\mathcal{P}_2(\mathcal{X}), \mathcal{W}_p)$ is referred to as the Wasserstein probability space (Villani et al., 2009).

Note that directly calculating $\mathcal{W}_p$ is computationally expensive, especially for high dimensional problems (Santambrogio, 2015). Consequently, the entropy-regularized Wasserstein distance (Cuturi, 2013) is proposed to approximate equation 1 by regularizing the original problem with an entropy term:

**Definition 1.** *The entropy-regularized Wasserstein distance is formally defined as:*

$$\mathcal{W}_{p,\varepsilon}(\mu, \nu) = \inf_{\pi \in \Pi(\mu, \nu)} \left[ \left( \int_{\mathcal{X} \times \mathcal{X}} \|\boldsymbol{x} - \boldsymbol{y}\|^p \, \mathrm{d}\pi(\boldsymbol{x}, \boldsymbol{y}) \right)^{\frac{1}{p}} + \varepsilon KL(\pi | \mu \otimes \nu) \right], \tag{2}$$

*where $\varepsilon > 0$ is a regularization coefficient, $\mu \otimes \nu$ denotes the product measure, i.e., $\mu \otimes \nu(\boldsymbol{x}, \boldsymbol{y}) = \mu(\boldsymbol{x})\nu(\boldsymbol{y})$, and $KL(\pi | \mu \otimes \nu)$ denotes the KL-divergence between $\pi$ and $\mu \otimes \nu$.*

Generally, the computational cost of $\mathcal{W}_{p,\varepsilon}$ is much lower than $\mathcal{W}_p$, and can be efficiently calculated with Sinkhorn algorithms (Cuturi, 2013). Without loss of generality, we fix $p = 2$ and abbreviate $\mathcal{W}_{2,\varepsilon} := \mathcal{W}_\varepsilon$ for ease of notion in the whole paper. According to Fenchel-Rockafellar theorem, the entropy-regularized Wasserstein problem $\mathcal{W}_\varepsilon$ equation 2 has an equivalent dual formulation, which is given as follows Peyré et al. (2017):

$$\mathcal{W}_\varepsilon(\mu, \nu) = \max_{f, g \in \mathcal{C}(\mathcal{X})} \langle \mu, f \rangle + \langle \nu, g \rangle - \varepsilon \left\langle \mu \otimes \nu, \exp\left(\frac{1}{\varepsilon}(f \oplus g - \mathrm{C})\right) - 1 \right\rangle, \tag{3}$$

where $C$ is the cost function in 2 and $f \oplus g$ is the tensor sum: $(x, y) \in \mathcal{X}^2 \mapsto f(x) + g(y)$. The maximizers $f_{\mu,\nu}$ and $g_{\mu,\nu}$ of 3 are called the $\mathcal{W}_\varepsilon$-potentials of $\mathcal{W}_\varepsilon(\mu, \nu)$. The following lemma states the optimality condition for the $\mathcal{W}_\varepsilon$-potentials:

**Lemma 1.** *(Optimality Cuturi (2013)) The $\mathcal{W}_\varepsilon$-potentials $(f_{\mu,\nu}, g_{\mu,\nu})$ exist and are unique $(\mu, \nu) - a.e.$ up to an additive constant (i.e. $\forall K \in \mathbb{R}, (f_{\mu,\nu} + K, g_{\mu,\nu} - K)$ is optimal). Moreover,*

$$\mathcal{W}_\varepsilon(\mu, \nu) = \langle \mu, f_{\mu,\nu} \rangle + \langle \nu, g_{\mu,\nu} \rangle. \tag{4}$$

We describe such the method in Appendix A for completeness Note that, although computationally more efficient than the $\mathcal{W}_p$ distance, the $\mathcal{W}_\varepsilon$ distance is not a true metric, as there exists $\mu \in \mathcal{P}_2(\mathcal{X})$ such that $\mathcal{W}_\varepsilon(\mu, \mu) \neq 0$ when $\varepsilon \neq 0$, which restricts the applicability of $\mathcal{W}_\varepsilon$. As a result, the following Sinkhorn divergence $\mathcal{S}_\varepsilon(\mu, \nu) : \mathcal{P}_2(\mathcal{X}) \times \mathcal{P}_2(\mathcal{X}) \to \mathbb{R}$ is proposed (Peyré et al., 2017):

**Definition 2.** *Sinkhorn divergence:*

$$\mathcal{S}_\varepsilon(\mu, \nu) = \mathcal{W}_\varepsilon(\mu, \nu) - \frac{1}{2} \left( \mathcal{W}_\varepsilon(\mu, \mu) + \mathcal{W}_\varepsilon(\nu, \nu) \right). \tag{5}$$

$\mathcal{S}_\varepsilon(\mu, \nu)$ is nonnegative, bi-convex thus a valid metric on $\mathcal{P}_2(\mathcal{X})$ and metricize the convergence in law. Actually $\mathcal{S}_\varepsilon(\mu, \nu)$ interpolates the Wasserstein distance ($\epsilon \to 0$) and the Maximum Mean Discrepancy ($\epsilon \to \infty$) (Feydy et al., 2019).

### 3.3 GRADIENT FLOWS

Consider an optimization problem over $\mathcal{P}_2(\mathcal{X})$:

$$\min_{\mu \in \mathcal{P}_2(\mathcal{X})} \mathcal{F}(\mu) := \mathcal{D}(\mu | \mu^*). \tag{6}$$

where $\mu^*$ is the target distribution, $\mathcal{D}$ is the divergence we choose. We consider now the problem of transporting mass from an initial distribution $\mu_0$ to a target distribution $\mu^*$, by finding a continuous probability path $\mu_t$ starting from $\mu_0 = \mu$ that converges to $\mu^*$ while decreasing $\mathcal{F}(\mu_t)$. To solve this optimization problem, one can consider a descent flow of $\mathcal{F}(\mu)$ in the Wasserstein space, which transports any initial distribution $\mu_0$ towards the target distribution $\mu^*$. Specifically, the descent flow of $\mathcal{F}(\mu)$ is described by the following continuity equation (Ambrosio et al. (2005), Villani et al. (2009), Santambrogio (2017)):

$$\frac{\partial \mu_t(x)}{\partial t} = -\nabla \cdot (\mu_t(x) \boldsymbol{v}_t(x)). \tag{7}$$

where $\boldsymbol{v}_{\mu_t} : \mathcal{X} \to \mathcal{X}$ is a velocity field that defines the direction of position transportation. To ensure a descent of $\mathcal{F}(\mu_t)$ over time $t$, the velocity field $\boldsymbol{v}_{\mu_t}$ should satisfy the following inequality ((Ambrosio et al., 2005)):

$$\frac{\mathrm{d}\mathcal{F}(\mu_t)}{\mathrm{d}t} = \int \langle \nabla \frac{\delta \mathcal{F}(\mu_t)}{\delta \mu}, \boldsymbol{v}_t \rangle \mathrm{d}\mu_t \leq 0. \tag{8}$$

A straightforward choice of $\boldsymbol{v}_t$ is $\boldsymbol{v}_t = -\nabla \frac{\delta \mathcal{F}(\mu_t)}{\delta \mu}$, which is actually the steepest descent direction of $\mathcal{F}(\mu_t)$. When we select this $\boldsymbol{v}_t$, we refer to the aforementioned continuous equation as the *Wasserstein gradient flow* of $\mathcal{F}$. We give the definition of the first variation in the appendix for the sake of completeness of the article.

## 4 METHODOLOGY

In this section, we first introduce the Sinkhorn Wasserstein gradient flow and investigate its convergence properties. Then, we develop our Neural Sinkhorn Gradient Flow model, which consists of a velocity field matching training procedure and a velocity field guided inference procedure. Moreover, we theoretically show that the mean-field limit of the empirical approximation used in the training procedure converges to the true velocity field of the Sinkhorn Wasserstein gradient flow.

### 4.1 SINKHORN WASSERSTEIN GRADIENT FLOW

Based on the definition of the Sinkhorn divergence, we construct our Sinkhorn objective $\mathcal{F}_\varepsilon(\cdot) = \mathcal{S}_\varepsilon(\cdot, \mu^*)$, where $\mu^*$ denotes the target distribution. The following theorem gives the first variation of the Sinkhorn objective.

**Theorem 1.** *([First variation of the Sinkhorn objective Luise et al. (2019)](#)) Let $\varepsilon > 0$. Let $(f_{\mu,\mu^*}, g_{\mu,\mu^*})$ be the $\mathcal{W}_\varepsilon$-potentials of $\mathcal{W}_\varepsilon(\mu, \mu^*)$ and $(f_{\mu,\mu}, g_{\mu,\mu})$ be the $\mathcal{W}_\varepsilon$-potentials of $\mathcal{W}_\varepsilon(\mu, \mu)$. The first variation of the Sinkhorn objective $\mathcal{F}_\varepsilon$ is*

$$\frac{\delta\mathcal{F}_\varepsilon}{\delta\mu} = f_{\mu,\mu^*} - f_{\mu,\mu}. \tag{9}$$

According to Theorem 1, we can construct the Sinkhorn Wasserstein gradient flow by setting the velocity field $\boldsymbol{v}_t$ in the continuity equation equation 7 as $\boldsymbol{v}_{\mu_t}^{\mathcal{F}_\varepsilon} = -\nabla\frac{\delta\mathcal{F}_\varepsilon(\mu_t)}{\delta\mu_t} = \nabla f_{\mu_t,\mu_t} - \nabla f_{\mu_t,\mu^*}$.

**Proposition 1.** *Consider the Sinkhorn Wasserstein gradient flow described by the following continuity equation:*

$$\frac{\partial\mu_t(\boldsymbol{x})}{\partial t} = -\nabla \cdot (\mu_t(\boldsymbol{x})(\nabla f_{\mu_t,\mu_t}(\boldsymbol{x}) - \nabla f_{\mu_t,\mu^*}(\boldsymbol{x}))). \tag{10}$$

*The following local descending property of $\mathcal{F}_\varepsilon$ holds:*

$$\frac{\mathrm{d}\mathcal{F}_\varepsilon(\mu_t)}{\mathrm{d}t} = -\int \|\nabla f_{\mu_t,\mu_t}(\boldsymbol{x}) - \nabla f_{\mu_t,\mu^*}(\boldsymbol{x})\|^2 \, \mathrm{d}\mu_t, \tag{11}$$

*where the r.h.s. equals 0 if and only if $\mu_t = \mu^*$.*

### 4.2 VELOCITY-FIELDS MATCHING

We now present our NSGF method, the core of which lies in training a neural network to approximate the time-varying velocity field $\boldsymbol{v}_{\mu_t}^{\mathcal{F}_\varepsilon}$ induced by Sinkhorn Wasserstein gradient flow. Given a target probability density path $\mu_t(x)$ and it's corresponding velocity field $\boldsymbol{v}_{\mu_t}^{\mathcal{S}_\varepsilon}$, which generates $\mu_t(x)$, we define the velocity field matching objective as follows:

$$\min_\theta \mathbf{E}_{t\sim[0,T],x\sim\mu_t}\left[\left\|\boldsymbol{v}^\theta(x,t) - \boldsymbol{v}_{\mu_t}^{\mathcal{S}_\varepsilon}(x)\right\|^2\right]. \tag{12}$$

To construct our algorithm, we utilize independently and identically distributed (i.i.d) samples denoted as $\{Y_i\}_{i=1}^n \in \mathbb{R}^d$, which are drawn from an unknown target distribution $\mu^*$ a common practice in the field of generative modeling. Given the current set of samples $\{\tilde{X}_i^t\}_{i=1}^n \sim \mu_t$, our method calculates the velocity field using the $\mathcal{W}_\varepsilon$-potentials (Lemma 1) $f_{\tilde{\mu}_t,\tilde{\mu}^*}$ and $f_{\tilde{\mu}_t,\tilde{\mu}_t}$ based on samples. Here, $\tilde{\mu}_t$ and $\tilde{\mu}^*$ represent discrete Dirac distributions. Note that these potentials defined on discrete measures can be calculated efficiently with first-order stochastic gradient descent methods such as SGD and ADAM (Bottou et al., 2018; Kingma & Ba, 2014). The corresponding finite sample velocity field approximation can be computed as follows:

$$\hat{\boldsymbol{v}}_{\tilde{\mu}_t}^{\mathcal{F}_\varepsilon}(\tilde{X}_i^t) = \nabla_{\tilde{X}_i^t} f_{\tilde{\mu}_t,\tilde{\mu}_t}(\tilde{X}_i^t) - \nabla_{\tilde{X}_i^t} f_{\tilde{\mu}_t,\tilde{\mu}^*}(\tilde{X}_i^t). \tag{13}$$

Subsequently, we derive the particle formulation corresponding to the flow formulation 10.

$$\mathrm{d}\tilde{X}_i^t = \hat{\boldsymbol{v}}_{\tilde{\mu}_t}^{\mathcal{F}_\varepsilon}\left(\tilde{X}_i^t\right)\mathrm{d}t, i = 1, 2, \cdots n. \tag{14}$$

In the following proposition, we investigate the mean-field limit of the particle set $\{\tilde{X}_i^t\}_{i=1,\cdots,M}$.

**Theorem 2.** *(Mean-field limits.) Suppose the empirical distribution $\tilde{\mu}_0$ of $M$ particles weakly converges to a distribution $\mu_0$ when $M \to \infty$. Then, the path of equation 14 starting from $\tilde{\mu}_0$ weakly converges to a solution of the following partial differential equation starting from $\mu_0$ when $M \to \infty$:*

$$\frac{\partial\mu_t(x)}{\partial t} = -\nabla \cdot (\mu_t(x)\nabla\frac{\delta\mathcal{F}_\varepsilon(\mu_t)}{\delta\mu_t}). \tag{15}$$

*which is actually the gradient flow of Sinkhorn divergence $\mathcal{F}_\varepsilon$ in the Wasserstein space.*

The following proposition shows that the goal of the velocity field matching objective equation 12 can be regarded as approximating the steepest local descent direction with neural networks.

**Proposition 2.** *(Steepest local descent direction.) Consider the infinitesimal transport $T(x) = x + \lambda\phi$. The Fréchet derivative under this particular perturbation,*

$$
\begin{aligned}
\frac{\mathbf{d}}{\mathbf{d}\lambda}\mathcal{F}_\varepsilon(T_\#\mu)|_{\lambda=0} &= \lim_{\lambda\to 0}\frac{\mathcal{F}_\varepsilon\left(T_\#\mu\right) - \mathcal{F}_\varepsilon\left(\mu\right)}{\lambda} \\
&= \int_{\mathcal{X}}\nabla f_{\mu,\mu^*}(\boldsymbol{x})\phi(\boldsymbol{x})d\mu - \int_{\mathcal{X}}\nabla f_{\mu,\mu}(\boldsymbol{x})\phi(\boldsymbol{x})\mathbf{d}\mu,
\end{aligned}
\tag{16}
$$

*and the steepest local descent direction is $\phi = \frac{\nabla f_{\mu,\mu^*}(\boldsymbol{x}) - f_{\mu,\mu}(\boldsymbol{x})}{\|\nabla f_{\mu,\mu^*}(\boldsymbol{x}) - f_{\mu,\mu}(\boldsymbol{x})\|}$.*

The velocity field matching training procedure is outlined in Algorithm 1. Considering the balance between expensive training costs and training quality, we opted to first build a trajectory pool of Sinkhorn gradient flow and then sample from it to construct the velocity field matching algorithm. Our method draws inspiration from experience replsay, a common technique in reinforcement learning, adapting it to enhance our model's effectiveness (Mnih et al., 2013; Silver et al., 2016). Once we calculate the time-varying velocity field $\hat{\boldsymbol{v}}^s_{\mu_t}(\tilde{X}^t_i)$, we can parameterize the velocity field using a straightforward regression method.

**Remark 1.** *In the discrete case, $\mathcal{W}_\varepsilon$-potentials 1 can be computed by a standard method in Genevay et al. (2016). In practice, we use the efficient implementation of the Sinkhorn algorithm with GPU acceleration from the GeomLoss package (Feydy et al., 2019).*

---

**Algorithm 1: Velocity field matching training**

---

**Input** : number of time steps $T$, batch size $n$, gradient flow step size $\eta > 0$, empirical or samplable distribution $\mu_0$ and $\mu^*$, neural network parameters $\theta$, optimizer step size $\gamma > 0$

`/* Build trajectory pool                                             */`
**while** *Building* **do**
  `/* Sample batches of size n i.i.d. from the datasets      */`
  $\tilde{X}^0_i \sim \mu_0, \quad \tilde{Y}_i \sim \mu^*, i = 1, 2, \cdots n.$
  **for** $t = 0, 1, \cdots T$ **do**
    calculate $f_{\tilde{\mu}_t,\tilde{\mu}_t}\left(\tilde{X}^t_i\right), f_{\tilde{\mu}_t,\tilde{\mu}^*}\left(\tilde{X}^t_i\right)$ using the GeomLoss package .
    $\hat{\boldsymbol{v}}^{\mathcal{F}_\epsilon}_{\mu_t}\left(\tilde{X}^t_i\right) = \nabla f_{\tilde{\mu}_t,\tilde{\mu}_t}\left(\tilde{X}^t_i\right) - \nabla f_{\tilde{\mu}_t,\tilde{\mu}^*}\left(\tilde{X}^t_i\right).$
    $\tilde{X}^{t+1}_i = \tilde{X}^t_i + \eta\hat{\boldsymbol{v}}^{\mathcal{F}_\epsilon}_t\left(\tilde{X}^t_i\right).$
    store all $\left(\tilde{X}^t_i, \hat{\boldsymbol{v}}^{\mathcal{F}_\epsilon}_t\left(\tilde{X}^t_i\right)\right)$ pair into the pool, $\quad i = 1, 2, \cdots n, \quad t = 0, 1, \cdots T.$

`/* velocity field matching                                           */`
**while** *Not convergence* **do**
  from trajectory pool sample pair $\left(\tilde{X}^t_i, \hat{\boldsymbol{v}}^{\mathcal{F}_\epsilon}_t\left(\tilde{X}^t_i\right)\right).$
  $\mathcal{L}(\theta) = \left\|\boldsymbol{v}^\theta(\tilde{X}^t_i, t) - \hat{\boldsymbol{v}}^{\mathcal{F}_\varepsilon}_{\mu_t}\left(\tilde{X}^t_i\right)\right\|^2,$
  $\theta \leftarrow \theta - \gamma\nabla_\theta\mathcal{L}(\theta).$
**Output:** $\theta$ parameterize the time-varying velocity field

---

Once obtained a feasible velocity field approximation $\boldsymbol{v}^\theta$, one can generate new samples by iteratively employing the explicit Euler discretization of the Equation 14 to drive the samples to the target. More details can be found in appendix D.Note that various other numerical schemes, such as the implicit Euler method (Platen & Bruti-Liberati, 2010) and higher-order Runge-Kutta methods (Butcher, 1964), can be employed. In this study, we opt for the first-order explicit Euler discretization method (Süli & Mayers, 2003) due to its simplicity and ease of implementation. We leave the exploration of higher-order algorithms for future research.

| Algorithm | 2-Wasserstein distance (10 steps) | | | | | 2-Wasserstein distance (100 steps) | | | | |
|---|---|---|---|---|---|---|---|---|---|---|
| | 8gaussians | 8gaussians-moons | moons | scurve | checkerboard | 8gaussians | 8gaussians-moons | moons | scurve | checkerboard |
| NSGF(ours) | **0.285** | **0.144** | **0.077** | **0.117** | **0.252** | 0.278 | **0.144** | **0.067** | **0.110** | **0.147** |
| JKO-Flow | 0.290 | 0.177 | 0.085 | 0.135 | 0.269 | 0.274 | 0.167 | 0.085 | 0.123 | 0.160 |
| EPT | 0.295 | 0.180 | 0.082 | 0.138 | 0.277 | 0.289 | 0.176 | 0.080 | 0.118 | 0.163 |
| OT-CFM | 0.289 | 0.173 | 0.088 | 0.149 | 0.253 | **0.269** | 0.165 | 0.078 | 0.127 | 0.159 |
| 1-RF | 0.427 | 0.294 | 0.107 | 0.169 | 0.396 | 0.415 | 0.293 | 0.099 | 0.136 | 0.166 |
| 2-RF | 0.428 | 0.311 | 0.125 | 0.171 | 0.421 | 0.430 | 0.311 | 0.121 | 0.136 | 0.170 |
| 3-RF | 0.421 | 0.298 | 0.110 | 0.170 | 0.413 | 0.414 | 0.297 | 0.103 | 0.140 | 0.170 |
| SI | 0.435 | 0.324 | 0.134 | 0.187 | 0.427 | 0.411 | 0.294 | 0.096 | 0.139 | 0.166 |
| FM | 0.423 | 0.292 | 0.111 | 0.171 | 0.417 | 0.415 | 0.290 | 0.097 | 0.135 | 0.165 |

Table 1: Comparison of neural gradient-flow-based methods and neural ODE-based diffusion models over five data sets. We provide the generated result in 10/100 Euler steps for Neural ODE Models, EPT, and our methods. The principle of steps in JKO-flow means backward Eulerian method steps (JKO steps).

## 5 EXPERIMENTS

We conducted an empirical investigation of the *Neural Sinkhorn Gradient Flow (NSGF)* method across a range of experiments. Initially, we demonstrate how NSGF guides the evolution and convergence of particles from the initial distribution toward the target distribution in 2D distribution experiments. Subsequently, we shift our focus to benchmarking on high-dimensional datasets, specifically the MNIST dataset (LeCun et al. (1998)) and the CIFAR-10 dataset (Krizhevsky et al. (2009)). Our method's adaptability to high-dimensional spaces is exemplified through experiments conducted on these datasets.

### 5.1 2D SIMULATE DATA

We assess the performance of various generative modeling models in low dimensions. Specifically, we conduct a comparative analysis between our method, NSGF, and several neural ODE-based diffusion models, including Flow Matching (FM; Lipman et al. (2023)), Rectified Flow (1,2,3-RF; Liu et al. (2023)), Optimal Transport Condition Flow Matching (OT-CFM; Tong et al. (2023); Pooladian et al. (2023)), Stochastic Interpolant (SI; Albergo & Vanden-Eijnden (2023)), and neural gradient-flow-based models such as JKO-Flow (Fan et al., 2022) and EPT (Gao et al., 2022a). Our evaluation involves learning 2D distributions adapted from Grathwohl et al. (2018), which include multiple modes.

Table 1 provides a comprehensive overview of our 2D experimental results, clearly illustrating the generalization capabilities of NSGF. Even when employing fewer steps. It is evident that neural gradient-flow-based models consistently outperform neural ODE-based diffusion models, particularly in low-step settings. This observation suggests that neural gradient-flow-based models generate more informative paths, enabling effective generation with a reduced number of steps. Furthermore, our results showcase the best performances among neural gradient-flow-based models, indicating that we have successfully introduced a lower error in approximating Wasserstein gradient flows. More complete details of the experiment can be found in the appendix E In the absence of specific additional assertions, we adopted Euler steps as the inference steps.

We present additional comparisons between neural ODE-based diffusion models and neural gradient-flow-based models, represented by NSGF and EPT, in Figure 1, 2, which illustrates the flow at different steps from $0$ to $T$. Our observations reveal that the velocity field induced by NSGF exhibits notably high-speed values right from the outset. This is attributed to the fact that NSGF follows the steepest descent direction within the probability space. In contrast, neural ODE-based diffusion models, particularly those based on stochastic interpolation, do not follow the steepest descent path in 2D experiments. Even with the proposed rectified flow method by Liu et al. (2023) to straighten the path, these methods still necessitate more steps to reach the desired outcome.

### 5.2 IMAGE BENCHMARK DATA

In this section, we illustrate the scalability of our algorithm to the high-dimensional setting by applying our methods on real image datasets where only samples from the target distribution are accessible.

**Generative modeling for MNIST** We evaluate NSGF on MNIST to show our generating ability. Figure 3 shows samples and their trajectories starting from Gaussian noise to target distribution and

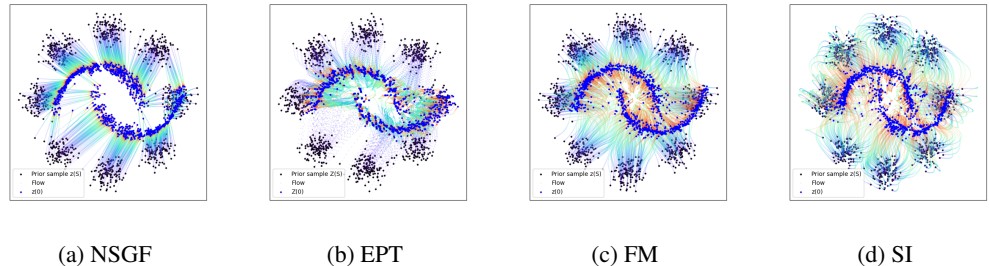

(a) NSGF     (b) EPT     (c) FM     (d) SI

Figure 1: Visualization results for 2D generated paths. We show different methods that drive the particle from the prior distribution (black) to the target distribution (blue). The color change of the flow shows the different number of steps (from blue to red means from $0$ to $T$).

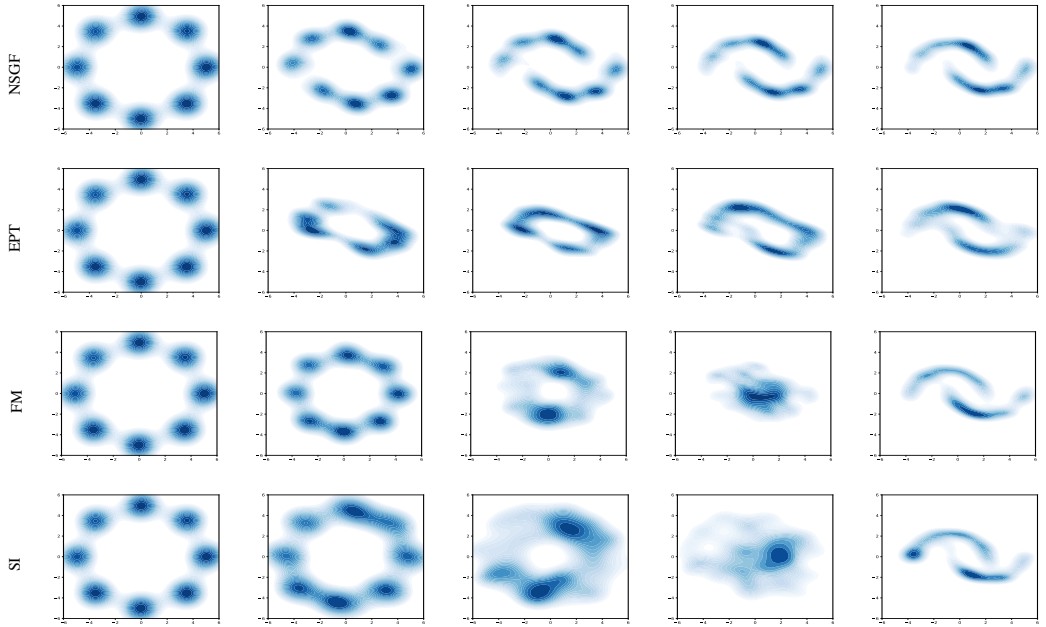

Figure 2: 2-Wasserstein Distance of the generated process utilizing neural ODE-based diffusion models and NSGF. The FM/SI methods reduce noise roughly linearly, while NSGF quickly recovers the target structure and progressively optimizes the details in subsequent steps which cause more meaningful interpolation between initial and target distributions.

demonstrates NSGF can approximate Sinkhorn gradient flow in image space empirically. We also provide sample quality using the standard Fréchet Inception Distance (FID) (Heusel et al., 2017) comparing with nonparametric gradient-flow-based methods SWGF (Sliced Wasserstein gradient flow, Liutkus et al. (2019)) and normalizing flows method SIG (Sliced iterative normalizing flows, Dai & Seljak (2020)) on appendix 3.

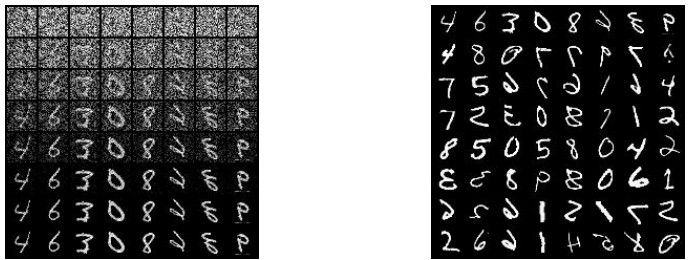

Figure 3: Trajectories and uncurated samples of our methods on MNIST

**Generative modeling for CIFAR-10** We report sample quality using the standard Fréchet Inception Distance (FID) (Heusel et al., 2017), Inception Score (IS) (Salimans et al., 2016) and compute

| Algorithm | CIFAR 10 | | |
|---|---|---|---|
| | IS(↑) | FID(↓) | NFE(↓) |
| NSGF(ours) | **7.56** | **21.6** | **100** |
| EPT(Gao et al., 2022a) | / | 46.63 | 10k |
| JKO-Flow(Fan et al., 2022) | 7.48 | 23.7 | >150 |
| DGGF(Heng et al., 2022) | / | 28.12 | 110 |
| OT-CFM(Tong et al., 2023) | / | 11.139 | 100 |
| FM(Lipman et al., 2023) | / | 6.35 | 142 |
| 1-RF(Liu et al., 2023) | **9.60** | **2.58** | 127 |
| 2-RF(Liu et al., 2023) | 9.24 | 3.36 | 110 |
| 3-RF(Liu et al., 2023) | 9.01 | 3.96 | **104** |
| SI(Albergo & Vanden-Eijnden, 2023) | / | 10.27 | / |

Table 2: Comparison of Neural Wasserstein gradient flow methods and Neural ODE-based diffusion models over CIFAR-10

cost using the number of function evaluations (NFE). These are all standard metrics throughout the literature.

Table 2 presents the results, including the Fréchet Inception Distance (FID), Inception Score (IS), and the number of function evaluations (NFE), comparing the empirical distribution generated by each algorithm with the target distribution. While our current implementation may not yet rival state-of-the-art methods, it demonstrates promising outcomes, particularly in terms of generating quality (FID), outperforming neural gradient-flow-based models (EPT, Gao et al. (2022a); JKO-Flow, Fan et al. (2022); DGGF,(LSIF-$\mathcal{X}^2$) Heng et al. (2022)) with fewer steps. It's essential to emphasize that this work represents an initial exploration of this particular model category and has not undergone optimization using common training techniques found in recent diffusion-based approaches. Such techniques include the use of exponential moving averages, truncations, learning rate warm-ups, and similar strategies. Furthermore, it's worth noting that training neural gradient-flow-based models like NSGF in high-dimensional spaces can be challenging. Balancing the optimization of per-step information with the limitations of the neural network's expressive power presents an intriguing research avenue that warrants further investigation. Figure 4 shows the trajectories of rectified flow (Liu et al., 2023) and NSGF. We still observe that NSGF quickly recovers the target structure and progressively optimizes the details in subsequent steps compared to the rectified flow.

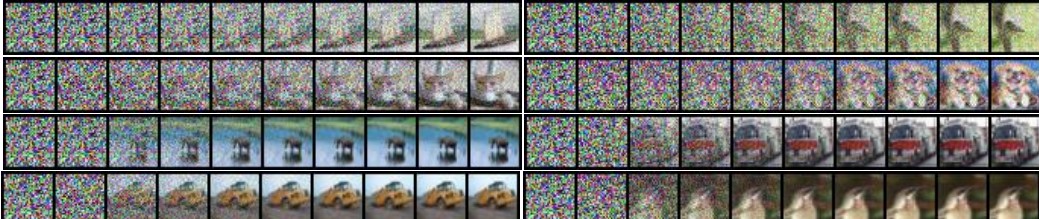

Figure 4: Trajectories comparison between the rectified flow and the NSGF model in CIFAR-10 task. The top two rows show the trajectories of rectified flow and the bottom two rows show the trajectories of the NSGF model. We can see NSGF model quickly recovers the target structure and progressively optimizes the details in subsequent steps

## 6 CONCLUSION

This paper delves into the realm of Wasserstein gradient flow w.r.t. the Sinkhorn divergence as an alternative to kernel methods. Our main investigation revolves around the Neural Sinkhorn Gradient Flow (NSGF) model, which introduces a parameterized velocity field that evolves over time in the Sinkhorn gradient flow. One noteworthy aspect of the NSGF is its efficient velocity field matching, which relies solely on samples from the target distribution for empirical approximations. The combination of rigorous theoretical foundations and empirical observations demonstrates that our approximations of the velocity field converge toward their true counterparts as the sample sizes grow. Through extensive empirical experiments on well-known datasets like MNIST and CIFAR-10, we provide further evidence supporting our theoretical claims and showcasing the NSGF's potential to surpass existing benchmarks for neural Wasserstein gradient flow. In conclusion, it becomes evident that the NSGF opens up new possibilities in the field of optimal transport, establishing a benchmark that will guide future research endeavors in this domain.

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
