# OpenReview forum: "Neural Sinkhorn Gradient Flow"
_ICLR.cc/2024/Conference — Submitted to ICLR 2024_

### Official Review · Reviewer_AcYB · 2023-10-18

**Soundness:** 3 good
**Presentation:** 2 fair
**Contribution:** 2 fair
**Rating:** 5
**Confidence:** 4

**Summary:**

The authors introduce Neural Sinkhorn gradient flow, which is a Wasserstein Gradient Flow wrt to the Sinkhorn divergence. The authors show that the velocity field can be calculated using the Sinkhorn potentials. This allows training a neural network approximating the velocity field. Furthermore, a mean field limit is established. The algorithm is evaluated on a toy example, MNIST image generation and CIFAR10 image generation.

**Strengths:**

The authors do a good job at explaining the underlying concepts of their algorithms. The maths is nicely done. The core idea is very neat and the cifar10 results seem to be good quantitatively wrt other gradient flow works.

**Weaknesses:**

1) The article is full with typos. Just to name a few: "piror", "Sinkhron", "Experimrnts", "speedest descent", question mark in the appendix and so on. Please fix those.

2) the authors write "We do not compare with extant neural WGF methods on MNIST because most of the neural WGF
methods only show generative power and trajectories on this dataset and lack the criteria to make
comparisons." There are several papers (also gradient flow based ones), which evaluate a FID on MNIST. Please provide it as well.

3) Also many of the MNIST digits appear flipped. Did the authors use data augmentation there? Also there seems to some slight noise present the generated MNIST digits.

4) Although the CIFAR10 value seems good, there are unfortunately no generated images provided. It is standard practice to sample many images in the appendix.

5) It is unclear what the trajectories show. Does it show the particle flow or the trained Neural Sinkhorn Gradient Flow?

6) The statement of theorem 2 is incorrect. I guess the authors do not want to sample the Euler scheme (eq 14) but the continuous gradient flow, otherwise the statement would need to depend on the step size $\eta$.

7) In the proof of Theorem 2: Please provide a proof (or reference) why the mean field limit exists. Or do you mean the gradient flow starting at $\mu_0$ with target $\mu$ (first two sentences).

8) Later in that proof: why does there exists a weakly convergent subsequence of $\mu_t^M$? Further, I cant find the definition of $U_{\mu}$.

9) The code is not runnable, as the model (or any checkpoints) are not provided.

10) From how I understood it, the learning of the velocity field is batched, i.e., one trains for different sets of $(z_i,x_i)$. Since the Sinkhorn dynamic describes an interacting particle system I dont see how this should be possible. To be more precise, one particle $\tilde{x}$ could be sent to $x_0$ in the first batch, but to a totally different particle $x_1$ in another one, depending on the drawn prior and target samples. Are the positions of the other particles also input to the neural network (i.e by putting them in the channels)? Please elaborate.

**Questions:**

See weaknesses section. Overall I really like the idea, but the weaknesses prevent me from giving a higher score. It seems like the paper was rushed and is currently not ready for publication. I am willing to raise my score, if the authors address these issues.

---

> ### Author Response · Authors · 2023-11-20
> **Response to Reviewer AcYB**
>
> Thank you for your appreciation of our idea, we will answer your questions one by one regarding these weaknesses/problems.
>
> **Answer to Weakness 1:** "Typos"
>
> We have fixed all the typos, thank you!
>
> **Answer to Weakness 2:** "There are several papers (also gradient flow-based ones), which evaluate a FID on MNIST. Please provide it as well."
>
> In fact, we are very willing to make quantitative comparisons with different methods, but due to limited time and computational resources, we did not replicate related algorithms on MNIST and test FID. The literature we reviewed that conducted MNIST experiments with neural WGF-based methods did not provide relevant data, including references [A] and [B]. However, we found that references [C], [G], and [H] evaluate an FID on MNIST. We have provided both our FID values and the FID values given in reference [C] for your reference. We are also very willing to add more comparisons if you could point us to those papers.
>
> **Answer to Weakness 3:** "Many of the MNIST digits appear flipped"
>
> We use data augmentation methods during training, including horizontal flipping on image datasets MNIST/CIFAR10. This is a common practice in the training of generative models like diffusion models [D]. In the revised version of our paper, we have provided in the appendix the results of retraining on datasets that were not subjected to data augmentation.
>
> **Answer to Weakness 4:** "Generated images provided"
>
> Due to space constraints, we have added sampling images in the Appendix.
>
> **Answer to Weakness 5:**  "It is unclear what the trajectories show"
>
> The trajectories show the Neural Sinkhorn Gradient Flow. In NSGF, a $32\times 32$ image can be understood as a particle in the flow. We have provided a clearer description of Figure 4 in the revised version.
>
> **Answer to Weakness 6:** "The statement of theorem 2"
>
> We have revised Equation 14 to an ODE (Ordinary Differential Equation) form in the amended version.
>
> **Answer to Weakness 7 and Weakness 8:** "The proof of Theorem 2"
>
> Apologies for the previous lack of clarity. We have re-examined and revised the proof of Theorem 2, and these modifications are included in the revised version of our paper (see Appendix for details). To prove that the empirical distribution $\tilde{\mu}^M_t$ evolving with equation 14 weakly converges to $\mu_t$ which is a solution of the equation 15 when $M\to \infty$ under the assumption $\\tilde{\mu}\_0^M \\rightharpoonup \\mu\_0$, we show that for arbitrary bounded and continuous $h$ and any $t$,
> $\\lim_{M\\to \\infty}\\mathbb{E}\_{\tilde{\mu}^M\_t}[h]\to\mathbb{E}\_{\mu\_t}[h]$.
> We show this property by Induction over the Continuum and analysis to infinitesimal of $\mathbb{E}_{\tilde{\mu}^M_t}[h]$ and $\\mathbb{E}\_{\\mu\_t}[h]$.
>
> **Answer to Weakness 9:** "The code is not runnable"
>
> We have added a readme and the missing neural network structure components to ensure the completeness of the code.
>
> **Answer to Weakness 10:** "Minibatch"
>
> The regression of vector fields with minibatches serves as an approximation to the potential vector fields of Sinkhorn gradient flows. This method effectively circumvents the problem of repeated pairing, which becomes more prominent as the batch size increases—a challenge commonly encountered in the training of most neural ODE/SDE-based and neural gradient-flow-based generative models, as highlighted in studies like [E], [F], and others. Interestingly, our experiments on toy datasets revealed that, even with small batch sizes, the outcomes of minibatch pairing training were on par with those achieved using larger batches.
> Furthermore, we can assure that when the samples from the initial and target distributions are identical, we can generate the exact same Sinkhorn gradient flow for training, a consistency that [I] and [J] do not guarantee.
>
> [A] Variational Wasserstein gradient flow, ICML'2022
> [B] Deep Generative Learning via Euler Particle Transport, MSML'2022
> [C] Deep Generative Learning via Variational Gradient Flow, ICML'2019
> [D] Denoising diffusion probabilistic models, NIPS'2020
> [E] Multisample Flow Matching: Straightening Flows with Minibatch Couplings, ICML'2023
> [F] Improving and Generalizing Flow-Based Generative Models with Minibatch Optimal Transport, ICML Workshop'2023
> [G] Sliced-Wasserstein Flows: Nonparametric Generative Modeling via Optimal Transport and Diffusions, ICML'2019
> [H] Sliced Iterative Normalizing Flows, ICML'2021
> [I] Flow Matching for Generative Modeling, ICLR'2023
> [J] Flow Straight and Fast: Learning to Generate and Transfer Data with Rectified Flow, ICLR'2023

---

> ### Comment · Reviewer_AcYB · 2023-11-20
> **Thanks for the rebuttal**
>
> While I think this paper has improved I still have some concerns.
>
> 1) Not all typos are fixed. In the appendix there are still several "Sinkhron"
>
> 2) Regarding 10) please comment on the U-net. As I understand your code (which now includes the model) this has 128 channels, which enables learning the particle dynamics. However, this would become very costly for more particles as the number of parameters in the U-net scales with the channels. So in order to get the mean field limit one would like to take number of particles to $\infty$, however this is probably very computationally expensive.
>
> 3) The cifar samples look good, could you also provide l2-nearest neighbors from the training set in order to show how your model generalizes.
>
> 4) can you please also upload the checkpoints of the u-nets? The memory requirements are not really feasible for me to train. If i understand correctly, then I think this will take a lot of velocity field/flow saves in the range of terabytes. (Is that correct? Then this should be mentioned in the paper)

---

> ### Author Response · Authors · 2023-11-23
> **Response to Reviewer AcYB**
>
> Thank you for your thorough review and valuable suggestions, particularly regarding paper revisions. Your insights have greatly enhanced the clarity and detail of our paper. We sincerely appreciate your guidance.
>
> 1. We apologize and have promptly rechecked and corrected the spelling errors.
>
> 2. Generally speaking, the choice of channel count in a U-net reflects a trade-off between the neural network's expressive capacity and the computational and storage costs. In our case, we use 128 channels, a common setting in generative models that employ U-net as their neural network architecture. It's foreseeable that expressing particle flows under mean field limit conditions requires a very deep neural network. However, we wish to emphasize that our demonstration of particle flows in the mean field limit as spatially continuous flows are to showcase our algorithm generating along paths that are more "information-rich". Furthermore, the theory related to mean field limits has also been developed in Bayesian sampling methods, especially in particle VI. Our insights are derived from advancements in this field.
>
> 3. Thank you for your suggestion. We have included a discussion on sample images and their nearest neighbors in the appendix of the revised version.
>
> 4.  Your understanding is correct. The volume of data stored during the first phase of our algorithm increases with the dimensionality of the dataset. For the MNIST/CIFAR-10 experiments, a considerable amount of storage space is needed to establish the trajectory pool during the first phase of the algorithm. Specifically, for CIFAR-10, setting the batch size to 128 and saving all minibatch Sinkhorn gradient flow trajectories while traversing the entire dataset requires about 115GB of storage space. In cases where storage space is limited, we recommend dynamically adding and removing trajectories in the trajectory pool to accommodate training needs. Identifying a more effective trade-off between training time and storage space utilization is a direction for future improvement. We will add a discussion on this in the updated version of our paper.   We note that the supplementary materials are subject to a 100 MB maximum file size limit. Unfortunately, even after compression into a ZIP format, our neural network's size significantly surpasses this limit. With a commitment to reproducibility and aiding the adoption of our approach, we intend to make the code publicly available following the publication of our paper. We try to use "anonymous.4open.science" but meet the same limit: "this exceeds GitHub's file size limit of 100.00 MB"

---

> > ### Comment · Reviewer_AcYB · 2023-11-23
> > **Thanks for the rebuttal**
> >
> > I still need to give the paper a proper re-read, but I am raising my score to 5 at least in light of the nice updates.

---

### Official Review · Reviewer_KKim · 2023-10-26

**Soundness:** 2 fair
**Presentation:** 2 fair
**Contribution:** 2 fair
**Rating:** 5
**Confidence:** 4

**Summary:**

This paper introduces a novel way to train generative models. The authors want to approximate the gradient flow in the Wasserstein space.  They want to approximate the vector field which transports the source distribution to the real-data empirical distribution while minimizing the Sinkhorn divergence. The authors showed the analytical form of the vector field when one considers the Sinkhorn divergence and then they explain how to learn this vector field with a neural network through the simulation of a probability path. They showed that their procedures recover the true probability path when the number of iid samples goes to infinity. Finally, they validate their proposed method on several image-generative tasks.

**Strengths:**

i) The motivation and the introduction are clear

ii) Regressing vector fields has been a recent and popular approach with many different applications in machine learning. The proposed approach is interesting and appears to be novel. The theoretical results also show that the proposed method has appealing properties.

iii) The authors also provided several experiments showing interesting results from their methods.

**Weaknesses:**

The first thing I would like to highlight is that I have checked the provided code. I see several inconsistencies and weaknesses between the provided code and the paper:

1. There are several differences in the empirical implementation between the paper and the code. In Appendix A, the authors state that they are computing the entropic potential through stochastic optimization algorithms [Genevay et al, 2016]. However, this is not what is done in practice according to the provided code. In practice, the authors compute the potential between mini-batches of samples, they sample a minibatch of cifar10 experiments, then sample a minibatch of the source Gaussian, and simulate the gradient flows between the two minibatches. This style of minibatch approximation induces a bias that should at least be mentioned in the main paper but also discussed. Indeed, the authors do not compute the true Sinkhorn divergence but a minibatch approximation of it; this approximation is slightly different than the one from [1,2] and that should be discussed. I understand the reason why the authors use this approach (decreasing the cost of this preprocessing step), but this is not what they say they do in Appendix A. In that regard, the paper is much closer to the minibatch optimal transport Flow Matching [Pooladian et al., Tong et al] and Appendix A deserves a major revision.

2. With the provided code, there are several insights that should be discussed in the paper. In the provided cifar experiments, the number of Gaussian samples used is 50000 samples. This number is extremely low to approximate the semi-discrete OT. Therefore, a discussion regarding the statistical performance of the method is needed in my opinion.

3. As your method requires the simulation of the probability path, I wonder about the training time between your method and the recent Flow Matching approaches which are simulation free.

4. There are many typos in the paper (including in titles: ie ExperimRnts, Notaions) that lead to poor clarity...

5. The experiments include two toy datasets (synthetic 2D and MNIST). I would like to know how the method performs on other big datasets (Flowers, CelebA) or on other tasks such as single-cell dynamics [4].

6. The related work on optimal transport is incomplete. Several works used the sliced Wasserstein distance to perform gradient flows [3].

[1] Learning Generative Models with Sinkhorn Divergences, Genevay et al, AISTATS 2018
[2] Learning with minibatch Wasserstein, Fatras et al, AISTATS 2020
[3] Sliced-Wasserstein Flows: Nonparametric Generative Modeling via Optimal Transport and Diffusions
[4] TrajectoryNet: A Dynamic Optimal Transport Network for Modeling Cellular Dynamics

**Questions:**

1. [Pooladian et al., Tong et al.] proved that when the minibatch increases, they get closer to the true optimal transport cost (W_2^2). The interest of their method is that they can rely on minibatches and learn the vector field from an unlimited number of minibatches. Could you follow a similar approach and simulate the gradient flow during training? While it would be an expensive step in training, it might improve the metrics on the different generative model experiments.

2. What is the performance of your method concerning the number of simulation steps (ie Euler integration and its learning rate)?

3. What is the time of the preprocessing step concerning the training time?

4. Could you compare your method with OT-CFM [Pooladian et al., Tong et al.] on the synthetic data? I am curious to compare the differences.

In my opinion, the mentioned weaknesses have to be revised and this paper should go under a major revision. I deeply think that the experimental section should better highlight what is done in practice and the theoretical section should mention the different biases (statistical and minibatch). Therefore, I recommend rejecting the current manuscript as it does not meet the ICLR acceptance bar.


----- EDIT POST REBUTTAL -----

I thank the authors for their answers. I have read the updated manuscript. While it is now better than before, I suggest they add a limitation section where they describe the different biases in their algorithm. I understand the motivations of the paper. Overall, I think that the manuscript deserves another round of reviews but I have decided to move my score to 5 as they have given good answers.

---

> ### Author Response · Authors · 2023-11-20
> **Response to Reviewer KKim**
>
> Thank you for your suggestions, we will answer your questions one by one regarding these weaknesses/problems.
>
> **Answer to Weakness 1:** "The differences in the empirical implementation between the paper and the code."
>
> We utilized minibatch sampling for both initial and target distribution pairs and calculated Sinkhorn potentials to construct the trajectory of the Sinkhorn gradient flow. Similar minibatch approximations for calculating the optimal transport plan or the corresponding Sinkhorn distance are noted in [A], [B], [C], and [D], which vary from our method of constructing the minibatch Sinkhorn gradient flow. The theoretical impact of minibatch use has been discussed in [A], and [B], [C] demonstrating through experiments that constructing interpolation flows using minibatch OT plans can train flow-based models with approximate OT paths. We plan to conduct further experiments in our future work. We recognize that minibatch was not explicitly mentioned in the main text, and have addressed this in the revised version (see Algorithm 1 for details). Furthermore, we have added a discussion about the minibatch approximation in Appendix A.
>
> **Answer to Weakness 2:** "Discussion regarding the statistical performance of the method"
>
> In our experiments, we indeed used significantly more than 50,000 Gaussian samples. As detailed in the revised version of our Algorithm 1, we clarified that our method comprises two distinct phases: building the trajectory pool and velocity field matching. The number of Gaussian samples utilized during training is determined in the first phase, while the quantity of trajectory samples used for training the neural velocity fields depends on the second phase. In the first stage, each iteration generates new Sinkhorn WGF paths using freshly resampled Gaussian samples (refer to `main_sinkhorn.py`), which are subsequently employed for vector field matching (see `main_minibatchOT.py`).
> It's important to note that the parameter of 50,000 Gaussian samples in the code originates from `evaluator.py`, which facilitates the inference process of the NSGF model. For our CIFAR10 experiment, we set this parameter to 50,000 to generate an equivalent number of images for standard FID value testing. We apologize for any confusion this may have caused and have provided detailed descriptions for each file in the supplementary materials of the revised version.
>
> **Answer to Weakness 3:** "Training time between your method and the recent Flow Matching approaches"
>
> Our algorithm takes longer in training time compared to simulation-free Flow Matching approaches [A]，[B], and [C] (we tested the time required to reach the same $W_2$ in 2D experiments, and the table is attached below), but since our model follows the steepest descent flow, we use fewer steps in the inference process to reduce generation time. We consider this a trade-off, which is discussed in our 2D simulations.
> In our understanding, 'simulation-free' in Flow Matching approaches means that any time snapshot in the Gaussian probability path can be obtained through random interpolation between noise distribution and the target distribution. In the training part of our method, our snapshots come from the discretization of Sinkhorn WGF, obtained through the iterative solving method (see Algorithm 1). In our algorithm, the main complexity of obtaining a snapshot at a specific time $t$ comes from solving the $\mathcal{W}_{\varepsilon}$-potential, with a complexity of $\mathcal{O}(tn^2)$, where $n$ represents the dimension, and $t$ represents the time. Notably, in OT-CFM [B], the authors perform OT matching on the minibatch sample pairs before the stochastic interpolation algorithm, with a complexity of $\mathcal{O}(n^3log(n))$.
> It's important to emphasize that although our training process requires more time, this does not impact our inference process. Our experiments demonstrate that we can use fewer steps, which reduces the time needed for generation.
>
> |methods|FM|OTCFM|NSGF|
> | ---- | ---- | ---- | -----|
> |time|119.11|145.94|1727.59|
> Table 1: Time taken by three comparison methods to achieve $W_2<0.2$ in the 8gaussians-moons task, unit: seconds.
>
> **Answer to Weakness 4:** "typos"
>
> We have fixed all the typos, thank you!
>
> **Answer to Weakness 5:** "big datasets (Flowers, CelebA) or on other tasks such as single-cell dynamics"
>
> Due to time and computational power limitations, we will construct experiments with big datasets and single-cell dynamics in our future work.
>
> **Answer to Weakness 6:** "The related work on optimal transport is incomplete"
>
> We have added new references on optimal transport, thanks!

---

> ### Author Response · Authors · 2023-11-20
> **Response to Reviewer KKim**
>
> **Answer to Question 1:** "Learn the vector field from an unlimited number of minibatches"
>
> Yes, we can follow a similar approach in [B] and [C].
> In our experiments, we chose to pre-construct the Sinkhorn WGF training set and perform the vector field matching algorithm. However, in fact, running `main_sinkhorn.py` repeatedly to build a trajectory pool and then using the stored pre-trained data for vector field matching is equivalent to simulating the gradient flow during training and learning the vector field from an unlimited number of minibatches. In our image experiments, considering the balance between expensive training costs and training quality, we opted to first build a trajectory pool of Sinkhorn WGF and then sample from it to construct the velocity field matching algorithm. This is more thoroughly explained in the revised version of Algorithm 1, hoping to eliminate any confusion regarding the code.
>
> **Answer to Question 2:** "What is the performance of your method concerning the number of simulation steps"
>
> According to Tables 1 and 2, we can observe that our algorithm achieves better generative effects with fewer simulation steps. Similarly, from Table 1, it can be seen that using more simulation steps can enhance the quality of our generation.
>
> **Answer to Question 3:** "1. What is the time of the preprocessing step concerning the training time?"
>
> The main complexity of obtaining a snapshot at a specific time $t$ comes from solving the $\mathcal{W}_{\varepsilon}$-potential, with a complexity of $\mathcal{O}(tn^2)$. In practical terms, we observed that the preprocessing step consumed a considerable amount of time. Therefore, in our image experiments, we chose to first build a trajectory pool and subsequently sample from it for constructing the velocity field matching algorithm. This approach reduces the time spent on the preprocessing step, thereby enhancing the overall efficiency of our algorithm's training process.
>
>
> **Answer to Question 4:** "Could you compare your method with OT-CFM [B, C] on the synthetic data?"
>
> We have already provided some quantitative comparison data for OT-CFM in Tables 1 and 2. Considering the limited length of the paper, we have provided some quantitative image results (similar to Figure 1) in the revised version.
>
> [A] Learning with minibatch Wasserstein: asymptotic and gradient properties, AISTATS’2020
> [B] Multisample Flow Matching: Straightening Flows with Minibatch Couplings, ICML'2023
> [C] Improving and Generalizing Flow-Based Generative Models with Minibatch Optimal Transport, ICML Workshop'2023
> [D] Learning Generative Models with Sinkhorn Divergences, AISTATS'2018

---

> ### Comment · Reviewer_KKim · 2023-11-20
> **Thank you for the rebuttal**
>
> Dear authors,
>
> Thank you for your rebuttal which I have read carefully.
>
> [Minibatch OT]: While the authors acknowledge the difference between the original submission and the original provided code, they did not make a careful discussion regarding the minibatch OT. I expect the authors to explain and discuss how it is related and how it differs from existing work [A,B,C]. Authors's approximation is a special case of minibatch OT where they only consider specific terms (ie in the original code, they computed OT between minibatches from one partition of the training dataset). Theoretically, it would correspond to a special Incomplete U-statistics. Therefore, I think that a detailed discussion and comparison to minibatch OT (and its bias) is needed and a remark is not enough.
>
> [Practical implementation] Thank you, Algorithm 1 is much clearer now and reflects what is done in practice. I highly recommend authors to update the methodology section in the main paper to incorpore this practical implementation and to discuss it in a limitation section.
>
> [Training time and motivation] Thank you for having run these experiments. In my opinion, they highlight a (known) limitation of the simulation-based approach like the proposed method. The proposed approach takes more than 10 times than other Flow Matching methods to converge. Indeed, the point of Flow Matching method is to avoid the computation of the interpolation samples that can be computed directly on the fly, resulting in a lower training time. It is the interest of these recent flow matching methods compared to simulation-based method. Furthermore, while authors do (minibatch) OT, it is also the case of (minibatch) OT-CFM [Tong et al., Poolidean et al.] and they do not require any simulation during training. Therefore, it is hard to understand the method interest to this recent line of work and that might require a revised motivation section.
>
> [Inference performance] While I understand authors's points regarding the inference, Flow Matching methods take at most 150 steps to converge towards SOTA FIDs on large datasets [Lipman et al.]. The differences between authors's method and Flow Matching method is 50 steps with a non-competitive FID. Therefore, I disagree with the authors that their method is competitive in that aspect.
>
> [Related work]: I would recommend to include a detailed discussion on minibatch OT and other gradient flow based OT method.
>
> Thank you for your answers regarding my questions. They have been addressed.
>
> I still think that some key elements are missing from the current manuscript (like the theoretical section which should mention the different biases). Therefore, I keep recommending to reject the current manuscript.

---

> > ### Author Response · Authors · 2023-11-23
> > **Response to Reviewer KKim**
> >
> > Thank you for your detailed review and insightful feedback, which have been instrumental in enhancing the clarity and depth of our paper. Your suggestions, especially on revisions, are greatly appreciated.
> >
> > [Minibatch OT]:
> > We respect your suggestion but hold some differing views. In our code, while we do use minibatch data to compute the $\mathcal{W}_{\varepsilon}$-potential for constructing the minibatch Sinkhorn gradient flow, we iterate through the entire dataset in a minibatch manner, rather than "only considering specific terms." In this aspect, our approach aligns with that of [A, B, C]. Our primary distinction from [B, C] lies in their methodology: after computing the minibatch OT plan, [B, C] proceed with random interpolation to construct paths (which requires both distributions to maintain Gaussian properties for geodesic paths, or approximate methods for distribution requirements), whereas we construct the Sinkhorn gradient flow. This point is discussed in our related work section.
> >
> > [Training time and motivation]:
> > Regarding our motivation, we observed that Wasserstein Gradient Flows (WGF) is a field with extensive research and broad applications. In recent years, neural gradient-flow-based models have shown potential in generative domains. Models based on WGF simulate WGF paths and, compared to neural SDE/ODE-based models utilizing Ornstein-Uhlenbeck processes or random linear interpolation, are richer in path information and can converge more quickly to the target distribution. These properties are desirable (this is also the reason [B, C] introduced OT properties to [G], though [B, C] only considered the OT plan for initial and target samples, still using random linear interpolation paths). Therefore, we believe that while neural gradient-flow-based models are not simulation-free training methods, they are still a worthwhile area of research. Reducing the computational cost and training time for simulating the Sinkhorn gradient flow is a direction for our future work.
> >
> > [Inference performance]:
> > We believe that there are many training techniques that could enhance the performance of the NSGF model [E, F]. In this paper, we focused on leveraging the properties of Sinkhorn gradient flow and velocity matching to construct the NSGF model. It is noteworthy that we have achieved results surpassing those of existing neural gradient-flow-based models. Research on further improving the NSGF model will be pursued in our future work.
> >
> > [Related work]:
> > We have discussed the neural gradient flow-based OT method in the introduction and also discussed the approaches of [B, C] in the related work section. Due to space constraints, we will include additional discussion in the appendix.
> >
> > [E] Improved Denoising Diffusion Probabilistic Models, ICML'2021
> > [F] Diffusion Models Beat GANs on Image Synthesis, NIPS'2021
> > [G] Flow Matching for Generative Modeling, ICLR'2023

---

### Official Review · Reviewer_Kh9H · 2023-10-29

**Soundness:** 3 good
**Presentation:** 2 fair
**Contribution:** 2 fair
**Rating:** 6
**Confidence:** 4

**Summary:**

Through a series of approximations (and at times, really, relaxations) the authors show that the Sinkhorn gradient flow from one measure to another can be learned.  They do this by first reducing their relaxed problem to a vector field matching problem, and then proposing a neural network-based Algorithm for matching the Sinkhorn-Wasserstein flow's vector field by a neural network (though no convergence/approximation guarantees are proven).
The problem is interesting, and its solution is sufficiently novel to merit publication.

**Strengths:**

The problem is natural to study, the results are mathematically correct, and the experiments are convincing.

**Weaknesses:**

While the paper is mathematically correct, it does not provide theoretical justification for one of its main components, namely showing that approximate vector field matching yields approximate solutions for all time $t$.  I feel that without this guarantee, there is a gap in the theoretical viability of this model.  Nevertheless, this is a minor point since the length of a conference paper does not allow one to treat every such point.

There are minor typos throughout.
* E.g. euclidean instead of Euclidean
* $lim$ instead of $\lim$ atop page 15 in the appendix
* The positive scalar $\delta$ is not defined in the proof of Theorem $1$
* In the statement of Lemma 3: "teh" should read "the"

Some references are obscure
* For The fact that $\mu + t\delta \mu$ converges weakly to $\mu$, perhaps it is worth simply noting that due to linearity of integration (wrt to the measure term).

**Questions:**

Can it be shown that approximate vector field matching yields approximate solutions for all time $t$?

---

> ### Author Response · Authors · 2023-11-20
> **Response to Reviewer Kh9H**
>
> Thank you for your appreciation of our work, we will answer your questions one by one regarding these weaknesses/problems.
>
> **Answer to Weakness 1 & Question 1:**  "Approximate vector field matching yields approximate solutions for all time $t$"
>
> It is challenging to rigorously analyze the error bounds of neural networks fitting vector fields at all times $t$, as it is related to properties like the smoothness of the flow and the expressive power of neural networks. Qualitatively speaking, as the number of samples used for training and the number of steps increase, the fitting error of the parameterized vector field for all times $t$ will decrease. We will reserve further developments for our future work.
>
> **Answer to Weakness 2:** "Typos"
>
> We have fixed all the typos, thank you!
>
> **Answer to Weakness 3:** "Some references are obscure"
>
> Apologies for the unclear referencing in proof of Theorem 1. We have revised this reference in the revised version of our paper.

---

### Official Review · Reviewer_KkYD · 2023-10-31

**Soundness:** 2 fair
**Presentation:** 3 good
**Contribution:** 2 fair
**Rating:** 5
**Confidence:** 4

**Summary:**

The paper under consideration deals with the standard generative modelling setup (image generation from noise). To solve this problem, the authors propose to model the gradient flow w.r.t. the Sinkhorn divergence. The paper utilizes an explicit (forward) Euler discretization scheme, i.e., given a distribution $\mu_t$ at the current time step $t$, the proposed method aims at finding the subsequent distribution $\mu_{t + 1}$ following the gradient of the Sinkhorn divergence at point $\mu_t$. The authors validate their methodology on toy 2D setups as well as standard image benchmarks (MNIST and CIFAR10).

**Post-rebuttal update:** I thank the authors for the detailed answer. The majority of my concerns are properly addressed. I rise my score. However, I still tend to reject the paper. Also I agree with reviewer KKim that minibatch OT approximation should be discussed more thorougly. Thank you.

**Strengths:**

To the best of my knowledge, the framework of the gradient flow w.r.t. Sinkhorn divergence for pure generative modelling has not yet been considered. This indicates that the paper is indeed bringing something novel to the ML community. At the same time, the idea of the Sinkhorn gradient flow has already arisen in previous research. In particular, [A] solves Sinkhorn barycenter problems by adjusting a generative distribution towards the barycenter distribution with the help of a procedure called “functional gradient descent” which is actually the discretization of the gradient flow w.r.t. the sum of Sinkhorn divergences to the target distributions. At the same time, it is worth mentioning, that [A] just simulates particles and does not build a generative model.
Regarding the other strengths of the paper, I would like to note the well-organized Experiments section.

[A] Sinkhorn Barycenter via Functional Gradient Descent, NeurIPS’2020

**Weaknesses:**

- Some theoretical results from the paper are known. For example, the statement of Theorem 1 could be found in [B] (eq. 26) or [C] (eq. 8).
- The quality of the code provided is not good. There is no README/or other instruction to run the code. There are imports of non-existing classes. So, there is no possibility of checking (at least, qualitatively) the provided experimental results.

From my point, the main weakness of the proposed paper is the limited methodological contribution. The authors simulate the particles of data following Sinkhorn divergence - as I already mentioned, this is not a super fresh idea. To make a generative model from these simulated trajectories, the authors simply solve the regression task to learn the local pushforward maps. And that is it. Combined with the fact, that the practical performance of the proposed approach is far from being SOTA in the generative modelling, the overall contribution of the paper seems for me to be limited.

**Questions:**

- My main question (and, probably, one of the main of my concerns) is regarding the proposed methodology. The authors propose to compute certain $\mathcal{W}_{\varepsilon}$ potentials (on discrete support of available samples) and then somehow take the gradients of these potentials w.r.t. the corresponding samples (eq. (13)). From the paper it is not clear how to compute the gradients, because the obtained potentials look like vectors of sample size shape, which are obtained through the iterations of the Sinkhorn algorithm. As I understand, in practice, the authors utilize SampleLoss from the geomloss package ([B]).  The outcome of this observation is that [B] should be properly cited when deriving the algorithm (section 4.2). I recommend authors explicitly use SampleLoss in the algorithm's listing. It will contribute to the clearness of what's going on.
- The vector field of the Sinkhorn gradient flow is estimated by empirical samples. It is not clear how well this sample estimate approximates the true vector field. This point should be clarified. Note, that Theorem 2 works only for mean-field limit.
- In the Introduction section, the authors consider a taxonomy of divergences used for gradient flow modelling, namely, "divergences [...] with the same support" and "divergences [...]  with possible different support". As I understand, the first class is about $f-$ divergences and the second class is about the other types (like Sinkhorn, MMD etc.). I have a question regarding the provided examples of works which deal with the former or the latter type of divergences. The fact is that the works [D], [E], [F], [G] deal with KL-divergence (or f-divergence) minimization. That is why I wonder why did the authors classify them as the second class.
- A good work regarding poor expressiveness of ICNNs is [H].
- What is the “ground” set ($\S$ 3.1, first line).
- Table 1. What are the differences between 1-RF, 2-RF and 3-RF methods?

[B] Interpolating between Optimal Transport and MMD using Sinkhorn Divergences, AISTATS’2019

[C] Sinkhorn Barycenters with Free Support via Frank-Wolfe Algorithm, NeurIPS’2019

[D] Large-scale wasserstein gradient flows. NeurIPS'2021

[E] Optimizing functionals on the space of probabilities with input convex neural networks. TMLR

[F]  Proximal optimal tranport modeling of population dynamics. AISTATS

[G] Variational wasserstein gradient flow. ICML

[H] Do Neural Optimal Transport Solvers Work? A Continuous Wasserstein-2 Benchmark. NeurIPS’2021.

---

> ### Author Response · Authors · 2023-11-20
> **Response 1 to Reviewer KkYD**
>
> Thank you for your suggestions, we will answer your questions one by one regarding these weaknesses/problems.
>
> **Answer to Weakness 1:**  “Some theoretical results from the paper are known”
>
> For the sake of completeness of our paper, we have included the theory of Sinkhorn WGF derivation in our methods section. Obviously, regarding the discussion of $\mathcal{W}_{\varepsilon}$-potential, we are not the first to do so, and we have added more relevant references to enhance the clarity of our paper. It is important to emphasize that our work presents an analysis of our discrete and continuous flows in mean-field limits, which is novel.
>
> **Answer to Weakness 2:** "The quality of the code provided is not good."
>
> Apologies that only the algorithm part was provided in the code. We have added a readme and the missing neural network structure components to ensure the completeness of the code in the latest version.
>
> **Answer to Weakness 3:** "Limited methodological contribution"
>
> We observed that existing neural SDE/ODE-based models necessitate multiple iterations through a neural network to produce high-quality samples. We noticed that the generative flows used in these models, such as reverse Ornstein-Uhlenbeck processes or linear flows constructed via random matching, tend to incur information loss. Therefore, exploring Wasserstein Gradient Flows (WGFs) that maintain more information by considering the steepest descent direction is a worthwhile direction. Simultaneously, we recognize that current neural gradient-flow-based models also face issues with information loss. Given this context, we considered leveraging the steepest descent property of Sinkhorn WGF to develop the NSGF model. Remarkably, by using a straightforward velocity field matching approach, we achieved generative quality surpassing existing neural gradient-flow-based models. Our experiments also demonstrated other interesting properties, such as improved generative paths and more meaningful interpolations between distributions, as seen in Figures 1 and 2. Notably, we are the first to construct a generative model based on Sinkhorn WGF, achieving results superior to other neural WGF-based works in fewer steps. We believe that further leveraging techniques from existing diffusion models (such as special neural network structures [D], or rapid sampling techniques [E]) could further enhance the capability of the NSGF model. This is a direction for our future work.
>
> [A] Sinkhorn Barycenter via Functional Gradient Descent, NeurIPS’2020
> [B] Interpolating between Optimal Transport and MMD using Sinkhorn Divergences, AISTATS’2019
> [C] Sinkhorn Barycenters with Free Support via Frank-Wolfe Algorithm, NeurIPS’2019
> [D] Diffusion Models Beat GANs on Image Synthesis,  NIPS'2021
> [E] DPM-Solver: A fast ODE solver for diffusion probabilistic model sampling in around 10 steps, NIPS'2022

---

> ### Author Response · Authors · 2023-11-20
> **Response 2 to Reviewer KkYD**
>
> **Answer to Question 1:**  "explicitly use SampleLoss in the algorithm's listing"
>
> We have added the detailed computation procedure of $\mathcal{W}_{\varepsilon}$-potentials in section 4.2 of the revised version.
>
> In the original paper, we only explained the calculation of $\mathcal{W}_{\varepsilon}$-potentials and the use of the geomloss package in the Appendix. We apologize for this affecting the completeness of the paper.
>
> **Answer to Question 2:** "How well does this sample estimate approximate the true vector field"
>
> In our paper, we concentrate on the mean-field limit situation and plan to address the analysis of the exact approximation error in our future work. It is important to note that conducting error analysis and theoretical proofs about the use of empirical sampling to approximate vector fields in flow-based models is generally challenging. For example, relevant discussions are not provided in references [F] and [H]. Even the analysis of sample complexity for Sinkhorn distance and Sinkhorn potential has gradually developed over time, following the introduction of these concepts.
>
> **Answer to Question 3:** "The taxonomy in Introduction section"
>
> Based on our understanding, papers [I] and [J] develop algorithms designed for a broad range of functionals, not limited to KL divergence. For instance, [I] uses the Free Energy Functional in the experiment section. [K] utilizes the Free Energy Functional and neural network parametrization techniques to construct functionals for WGF and JKO algorithms. The Free Energy Functional relies solely on a single distribution and does not require the “same support” condition, leading us to categorize it under the second class, “With Possible Different Supports.” We acknowledge that our description of these algorithms lacked clarity, and we have made revisions to our revised version. Regarding [L], our reference in the second paragraph was initially to illustrate the “poor expressiveness” of ICNNs. We have modified this reference in the revised version to address any ambiguity it may have caused.
>
> **Answer to Question 4:**  "A good work regarding poor expressiveness of ICNNs"
>
> We have added that reference, thanks!
>
> **Answer to Question 5:** "What is the “ground” set"
>
> The "ground set" is simply the underlying set or space under consideration.
>
> **Answer to Question 6:** "What are the differences between 1-RF, 2-RF and 3-RF methods"
>
> Differences between 1-RF, 2-RF, and 3-RF methods depend on the rectified flow technique which recursively applies rectification to obtain a sequence of flows with increasingly straight paths. As the authors show in their experiments, although the rectified flow method produces straighter paths in favor of reducing the number of sampling steps, the accumulation of errors in recursive training may reduce the quality of generation. More details can be found in [M].
>
> [F] Flow Matching for Generative Modeling, ICLR'2023
> [H] Flow Straight and Fast: Learning to Generate and Transfer Data with Rectified Flow, ICLR'2023
> [I] Large-scale Wasserstein gradient flows, NeurIPS'2021
> [J] Optimizing functionals on the space of probabilities with input convex neural networks, TMLR
> [K] Proximal optimal transport modeling of population dynamics. AISTATS
> [L] Variational wasserstein gradient flow, ICML'2021
> [M] Do Neural Optimal Transport Solvers Work? A Continuous Wasserstein-2 Benchmark, NeurIPS’2021.

---

### Official Review · Reviewer_MEFG · 2023-11-05

**Soundness:** 3 good
**Presentation:** 3 good
**Contribution:** 3 good
**Rating:** 6
**Confidence:** 3

**Summary:**

This paper introduces the idea of learning a time-dependent velocity field of the Sinkhorn Wasserstein gradient flow from samples from the target distribution to calculate the empirical velocity field approximations. The paper supports its claim by showing that the mean-field limit of this process recovers the true Sinkhorn Wasserstein gradient flow. They also validated the process with some empirical studies.

**Strengths:**

The paper is well written and easy to follow. The proofs and arguments in the appendix are well-typed out and clear.  There are some nice diagrams in the empirical section to supports the claim the authors are making.

**Weaknesses:**

I think the experiments could be more extensive. One thing about this method is to investigate the number of samples needed. effectively learn the velocity field. This is one important experiment missing as is remains unclear how sample-efficient the proposed method is. It would also make the paper more completing if the method is applied to generative models that output discrete random variable like binary mnist or even language modelling.

**Questions:**

One possible question is what happens if we change the source distribution to be closer to the target distribution like it was from a generator how would the method perform there. Another question is to better understand the sample complexity of the method as the current method may not be sample efficient due to the empirical distribution being approximated using the samples.

---

> ### Author Response · Authors · 2023-11-20
> **Response to Reviewer MEFG**
>
> Thank you for your appreciation of our work, we will answer your questions one by one regarding these weaknesses/problems.
>
> **Answer to Weaknesses 1 & Question 2:**  “Sample complexity of the method”
>
> Note that the NSGF model is gradient-flow-based, and the training of NSGF can be divided into two stages (see Algorithm 1 for details). In the first stage, NSGF constructs trajectories simulating Sinkhorn WGF using target points and source points and stores trajectories in the trajectory pool. In the second stage, we sample from the trajectory pool for velocity matching algorithms instead of using the source and target points. A similar procedure is widely used in flow-based models such as [A], and [B]. As a result, our method has a similar sample complexity to existing flow-based models. We have added relevant experiments in our 2D experiments, see the table below for details:
>
> |methods|FM|OTCFM|NSGF|
> | ---- | ---- | ---- | -----|
> |number of samples|1.23*e6|8.70*e5|7.68*e5|
> Table 1: Trajectory samples were taken by three comparison methods to achieve $W_2<0.2$ in the 8gaussians-moons task.
>
> **Answer to Weaknesses2:** "Generative models that output discrete random variable"
>
> Thank you for discussing the application of the NSGF model in generative tasks involving discrete random variables. Although our current focus is on simulating Sinkhorn WGF in continuous state spaces, such as for real-valued images, we recognize the potential for extending the NSGF model to discrete state spaces. This extension would follow methodologies similar to those found in [C] and [D]. Exploring the generative capability of the NSGF model in discrete state spaces is an interesting extension, and we will investigate this in our future work.
>
> **Answer to Question 1:** "Change the source distribution to be closer to the target distribution"
>
> The NSGF model is designed to simulate Sinkhorn WGF from a source distribution to a target distribution. If the source distribution is closer to the target distribution, Sinkhorn WGF will converge more quickly. This allows us to use fewer steps and lower computational costs during both the training and inference processes of NSGF. [E] considers using gradient-flow-based methods to refine the samples from a generator, which improves the metrics of the generated images. The NSGF model can also be used to achieve similar improvement, and this will be a focus of our future work.
>
> [A] Flow Matching for Generative Modeling, ICLR'2023
> [B] Flow Straight and Fast: Learning to Generate and Transfer Data with Rectified Flow, ICLR'2023
> [C] Structured Denoising Diffusion Models in Discrete State-Spaces, NIPS'2021
> [D] Argmax Flows and Multinomial Diffusion: Learning Categorical Distributions, NIPS'2021
> [E] Refining Deep Generative Models via Discriminator Gradient Flow, ICLR'2021

---

> > ### Comment · Reviewer_MEFG · 2023-11-20
> > **Response to authros**
> >
> > Thank you for the thoughtful replies
> >
> > Reply to the Answer to Weaknesses 1 & Question 2
> >
> > I appreciate the table provided and the comment about how the presented method is similar to others in the literature.  My big issue with this method's sample complexity would be understanding how the sample complexity grows with the dimensionality of the data since you are using the empirical distribution. It might be convincing if a similar table for mnist could be made.

---

> > > ### Author Response · Authors · 2023-11-23
> > > **Response to Reviewer MEFG**
> > >
> > > Thank you once again for your diligent review and sincere feedback, which are of great significance in improving the quality of our paper and guiding our future work. Due to current limitations in time and computational resources, we are unable to provide comparative experiments on MNIST at this moment. We plan to explore this further in our future work.

---

### Meta-Review · Area_Chair_J54j · 2023-12-07

**Metareview:**

The paper propose to learn a vector field of the Sinkhorn divergence gradient flow and apply this vector field for generative modeling by using an explicit (forward) Euler discretization scheme.

Reviewers have found that the paper introduces some new ideas with nice theoretical supports. However, while some concerns have been addressed during rebuttal, most of them still believe that the paper still need to address some issues before acceptance (eg discussion/role of the minibatch).

**Justification For Why Not Higher Score:**

Concerns of the reviewers have not been fully addressed regarding the role of minibatches or the discrepancy between theory and practice.

**Justification For Why Not Lower Score:**

na

---

### Decision · Program_Chairs · 2024-01-16

Reject